# Emerging Role of *ODC1* in Neurodevelopmental Disorders and Brain Development

**DOI:** 10.3390/genes12040470

**Published:** 2021-03-25

**Authors:** Jeremy W. Prokop, Caleb P. Bupp, Austin Frisch, Stephanie M. Bilinovich, Daniel B. Campbell, Daniel Vogt, Chad R. Schultz, Katie L. Uhl, Elizabeth VanSickle, Surender Rajasekaran, André S. Bachmann

**Affiliations:** 1Department of Pediatrics and Human Development, Michigan State University, Grand Rapids, MI 49503, USA; caleb.bupp@spectrumhealth.org (C.P.B.); frischau@msu.edu (A.F.); bilinovi@msu.edu (S.M.B.); campb971@msu.edu (D.B.C.); vogtdan2@msu.edu (D.V.); schul620@mail.msu.edu (C.R.S.); uhlkatie@msu.edu (K.L.U.); Surender.Rajasekaran@spectrumhealth.org (S.R.); 2Department of Pharmacology and Toxicology, Michigan State University, East Lansing, MI 48824, USA; 3Center for Research in Autism, Intellectual, and Other Neurodevelopmental Disabilities, Michigan State University, East Lansing, MI 48824, USA; 4Spectrum Health Medical Genetics, Grand Rapids, MI 49503, USA; Elizabeth.vansickle@helendevoschildrens.org; 5Neuroscience Program, Michigan State University, East Lansing, MI 48824, USA; 6Pediatric Intensive Care Unit, Helen DeVos Children’s Hospital, Grand Rapids, MI 49503, USA; 7Office of Research, Spectrum Health, Grand Rapids, MI 49503, USA

**Keywords:** neural development, polyamine, ornithine decarboxylase, rs138359527 (NP_002530.1:p.Gly84Arg), variant data integration

## Abstract

Ornithine decarboxylase 1 (*ODC1* gene) has been linked through gain-of-function variants to a rare disease featuring developmental delay, alopecia, macrocephaly, and structural brain anomalies. *ODC1* has been linked to additional diseases like cancer, with growing evidence for neurological contributions to schizophrenia, mood disorders, anxiety, epilepsy, learning, and suicidal behavior. The evidence of *ODC1* connection to neural disorders highlights the need for a systematic analysis of *ODC1* genotype-to-phenotype associations. An analysis of variants from ClinVar, Geno2MP, TOPMed, gnomAD, and COSMIC revealed an intellectual disability and seizure connected loss-of-function variant, ODC G84R (rs138359527, NC_000002.12:g.10444500C > T). The missense variant is found in ~1% of South Asian individuals and results in 2.5-fold decrease in enzyme function. Expression quantitative trait loci (eQTLs) reveal multiple functionally annotated, non-coding variants regulating *ODC1* that associate with psychiatric/neurological phenotypes. Further dissection of RNA-Seq during fetal brain development and within cerebral organoids showed an association of *ODC1* expression with cell proliferation of neural progenitor cells, suggesting gain-of-function variants with neural over-proliferation and loss-of-function variants with neural depletion. The linkage from the expression data of *ODC1* in early neural progenitor proliferation to phenotypes of neurodevelopmental delay and to the connection of polyamine metabolites in brain function establish *ODC1* as a bona fide neurodevelopmental disorder gene.

## 1. Introduction

The ornithine decarboxylase 1 (*ODC1*) gene codes for a sentinel rate-limiting enzyme (ODC) that contributes to the production of polyamines (putrescine, spermidine, spermine) by the decarboxylation of ornithine to putrescine [1,2]. ODC and polyamines have been extensively studied in cancer and are upregulated in many hyperproliferative tissues that form tumors. The *ODC1* and other polyamine enzyme genes are dysregulated in MYC-amplified tumors such as neuroblastomas [3,4,5]. Genetic polymorphisms within *ODC1* are associated with outcomes in colorectal [6,7], gastric [8], breast [9], and prostate [10] cancers. Suppression of polyamines, for example, by the inhibition of ODC with α-difluoromethylornithine (DFMO) has been explored as a therapeutic strategy in a variety of cancers including neuroblastoma [11,12,13,14]. In pediatric neuroblastoma patients, DFMO has been demonstrated to be well tolerated in doses of up to 1500 mg/m^2^ twice daily for 21 day cycles as well as for maintenance therapy for two years with a dose of 750 ± 250 mg/m^2^ twice daily [14,15]. Administration of DFMO during pregnancy has been correlated to fetal development, namely, the alteration of skeletal variations with high levels of preimplantation issues at high dosage in material water, ranking it as a category C drug [16]. The combination of known therapeutic intervention with stratifying polymorphisms is an ideal combination for pharmacogenomic potential.

Since 2005, *ODC1* has been suggested to have human genomic connections to medical conditions, namely androgenetic alopecia [17]. Loss of *ODC1* in homozygous knockout mice are complete penetrance lethal, while heterozygous mice have significant decreases in body mass and fat content [18] (Appendix A). Polymorphisms around the *ODC1* gene are associated with multiple phenotypic traits through GWAS and PheWAS [19] (Appendix A) such as height (*p*-value 1 × 10^−45^), educational attainment (1 × 10^−25^), highest math class taken (2 × 10^−20^), prostate cancer (6 × 10^−17^), heel bone mineral density (2 × 10^−14^, breast cancer (4 × 10^−10^), neuroticism (4 × 10^−10^), and worrier/anxious feelings (FDR 8 × 10^−10^). The associations with neurological phenotypes, in addition to the expected cancer and bone growth phenotypes, speak to a need to establish the mechanisms and direct connection of these variants to changes in *ODC1* biology in multiple organ systems through systematic analysis.

ODC activity and polyamine production play key roles during embryonic development from implantation to fetal status, long suggesting potential genetic syndromes related to *ODC1* [20]. The regulation of polyamines and ODC activity is critical. This importance can be highlighted by the evolved role of ODC antizymes and antizyme inhibitors [21]. The ODC antizymes inhibit ODC, and its functional role is highly dependent on the level of polyamines in the cell [22,23,24]. The unique frameshift regulation of antizyme and role in degradation of ODC makes the antizyme system a novel regulatory component [25]. Adding to the complexity of this regulatory system is an antizyme inhibitor, a homologous *ODC1* gene that can inhibit the antizyme from ODC regulation [26,27].

The polyamine metabolite putrescine is involved in the production of GABA and thus involved in GABAergic interneurons [28]. Over production of putrescine in the brain has been connected to seizures and spatial learning [29]. In 2019, it was shown that polyamine metabolite processing disrupted by loss-of-function variants in deoxyhypusine synthase (DHPS) is also associated with global developmental delay and multiple neurological phenotypes [30]. These establish that the downstream products of ODC are related to neurons, yet do not directly connect ODC to neural function. Recently, two groups (including ours) have shown evidence for a new autosomal recessive *ODC1*-related syndrome in five patients, having disruptive gain-of-function variants within the C-terminus of the protein (OMIM # 619075 for BACHMANN-BUPP SYNDROME; BABS) [31,32,33]. These patients feature developmental delay, alopecia, macrocephaly, and varied structural brain anomalies, with noted elevation of putrescine. Newly diagnosed, unpublished, patients with *ODC1*-associated syndromes continue to help expand the phenotype. These patients, along with the growing evidence of GWAS and PheWAS, confirm the need for a renewed analysis of *ODC1* genotype-to-phenotype associations in human health and disease, with a particular focus on neurological roles. Within this study, we performed a deep assessment of *ODC1*-associated genetics, revealing multiple lines of support for *ODC1* genetics connected to neural development.

## 2. Materials and Methods

### 2.1. Coding Variant Screening

Sequences for *ODC1* were extracted from NCBI Orthologs, trimmed for open reading frames using TransDecoder [34] and aligned using codon based ClustalW [35]. Codon selection, sliding window conservation scores, and motif analysis are described in our previous work [36]. Post translational modifications were curated from UniProt, ELM (elm.eu.org/, accessed on 27 June 2020), or the Cuckoo based tools (gps.biocuckoo.cn/, accessed on 27 June 2020). Variants for *ODC1* were extracted from ClinVar [37], Geno2MP (geno2mp.gs.washington.edu), TOPMed (bravo.sph.umich.edu/freeze5/hg38/), gnomAD [38], or COSMIC [39] all on 27 June 2020. Each missense variant was analyzed with PolyPhen2 (probably = 1, possibly = 0.5), Provean (Deleterious = 1), SIFT (Damaging = 1), align-GVGD (Class C65/55 = 1, C45/35 = 0.5, C25/15 = 0.25), and our conservation score (max = 2) with a summed max variant score of 6. This variant score was multiplied by the 21-codon conservation and gnomAD allele frequency for a combined impact score. This combined score allows for ranking of variants that are predicted to impact function from multiple tools, prioritizing variants. Open Targets Genetics [19] (genetics.opentargets.org/, accessed on 27 June 2020) was used to extract phenotypes of GWAS and PheWAS of missense variants.

### 2.2. Protein Modeling and Dynamics

The ODC protein (UniProt accession P11926) dimer was homology modeled within YASARA [40], merging PDB files 2ON3, 7ODC, 2OO0, 1D7K, 4ZGY, 5BWA. Within YASARA, the G84R was created, followed by energy minimizations. Both wild type and G84R ODC were analyzed for a total of 20 nanoseconds with molecular dynamic simulations using AMBER14 force field [41] with explicit water. Analysis of molecular dynamic trajectory coordinates was done with md_analyze and md_analyzeres macros. Evolution analysis from our calculated conservation score was combined with the protein model by coloring amino acids manually or by integrating the protein alignment into ConSurf tools [42].

### 2.3. Ornithine Decarboxylase (ODC) Enzyme Assay

Wild type and mutant G84R ODC enzymes were expressed in *E. coli* and purified by Ray Biotech, Inc. To compare the enzyme activity of WT vs. G84R ODC, 100 ng of purified ODC enzymes were incubated with 200 μL of assay mix containing 6.25 mM Tris HCl (pH7.5), 100 μM l-ornithine, 50 μM pyridoxal-5-phosphate, 1.56 mM dithiothreitol (DTT), and 0.1 μCi [1-14C] l-ornithine (American Radiolabeled Chemicals, Inc., St. Louis, MO, USA; specific activity 55 mCi/mmol) in a microcentrifuge tube. The microcentrifuge tubes were then placed into scintillation vials containing a piece of filter paper saturated with 200 μL 0.1 M NaOH to capture the release of radiolabeled carbon dioxide. The samples were incubated in a 37 °C incubator while shaking for 30 min. The enzymatic reaction was stopped by adding 250 μL of 5 M sulfuric acid to each sample and incubating at 37 °C while shaking for 30 min. The microcentrifuge tubes were removed from the scintillation vials and 5 mL of scintillation fluid was added. Disintegrations per minute of each sample was measured using a TriCarb liquid scintillation counter (Perkin Elmer, Waltham, MA, USA). The specific ODC activity is expressed as nmol CO_2_/min/mg protein.

### 2.4. Expression Analysis

Expression of *ODC1* was extracted from GTEx [43], Human Protein Atlas [44], and various single cell expression databases including PangloaDB [45], Tabula Muris [46], and the Allan Brain Atlas Developmental Transcriptome [47]. Identified genes correlating with *ODC1* in the Allan Brain Atlas Developmental Transcriptome were assessed for enrichment of gene ontology and protein–protein interactions with STRING [48] or for connection to phenotype terms by text extraction of records within ClinVar [37]. Mouse developmental staining of *Odc1* was extracted from Genepaint [49].

### 2.5. Noncoding Variant Screening

Both expression and splicing quantitative trait loci (eQTL/sQTL) were extracted from GTEx v8 [43] for *ODC1* in June 2020. All SNPs in linkage disequilibrium (LD) with *ODC1* eQTLs or sQTLs were imputed for 0.8R^2^ correlation with SNiPA tools (snipa.helmholtz-muenchen.de/snipa3/?task=proxy_search, accessed on 27 June 2020) for all phase3 1000 genome populations. The variant list was then parsed for functional information using SNPnexus [50] to extract CADD scores and ENCODE/Roadmap Epigenomics overlapping peaks. Using the SNP coordinates and the USCS genome browser, GeneHancer [51], Switchgear TSS, ENCODE TF binding, and conserved transcription factor binding annotations were extracted. RegulomeDB [52] and SNP2TFBS [53] were run on the variant list to determine if the change impacted transcription binding. A compiled noncoding variant impact score was generated by summing the number of overlapping peaks from ENCODE, Roadmap, RegulomeDB, and GeneHancer and multiplying by CADD score. This score allows for the rapid screening of any variants where epigenetic knowledge is known and where there might be overlapping signals. Linkage disequilibrium structures were computed for high scored SNPs using SNiPA tools. Structural looping data for SNPs were obtained from Yue lab Hi-C browser tools (promoter.bx.psu.edu/hi-c/, accessed on 27 June 2020). All SNPs were assessed for phenotype connections through Geno2MP and ieu open GWAS project (gwas.mrcieu.ac.uk, accessed on 27 June 2020).

### 2.6. Induced Pluripotent Stem Cell (iPSC) Based Organoid RNA-Seq

Human induced pluripotent stem cell line AICS-0011 (Allen Brain Atlas) was cultured on StemAdhere (Nucleus Biologics) in mTeSR1 media (StemCell Technology) at 37 °C with 5% CO_2_. The line was differentiated to cerebral organoids using the STEMdiff™ Cerebral Organoid Kit (STEMCELL Technologies, #08570) as detailed before [54]. Single-cell suspensions of cerebral organoids were prepared by incubation with LiberaseTM (0.03 ug in 1 mL basal 2 media) for 10 min at 37 °C. After washing in Phosphate Buffered Saline (PBS) and filtration through a 40 µM cell strainer, dead cells were removed using the EasySep Dead Cell Removal Kit (StemCell Technology). Cells were resuspended in PBS + 0.04% bovine serum albumin (BSA) prior to library preparation and sequencing using the 10x Genomics Kit at the Van Andel Research Institute Genomics Core. Samples were sequenced on the Illumina HiSeq6000. For data analysis, Alevin [55] was used for demultiplexing and alignment to transcriptome, and Seurat [56] for cell cluster analysis. Data are available within BioProject PRJNA640661.

## 3. Results

### 3.1. Human Variation Screen

Recent insights from Bupp et al. and Rodan et al. [31,32] revealed a phenotype characterized by multiple abnormalities, but developmental delay and neuromuscular abnormalities predominated. Therefore, we performed a deep genomic assessment of ODC1 from missense variants (Figure 1) to noncoding variants (Figure 2). Using 220 open reading frame vertebrate sequences of ODC1, we developed conservation and codon selection level insight for all amino acids of the gene. Placing the per residue scores on a 21-codon sliding window (10 codons before, the site, and 10 codons after), we calculated highly conserved motifs within the gene (Figure 1A). Using a merged ODC protein structure (PDB accession codes 2ON3, 7ODC, 2OO0, 1D7K, 4ZGY, 5BWA), we mapped the six top motifs, all of which contribute to either the active site of the enzyme or to the protein dimerization (Appendix A). Integrated UniProt information for the protein active site along with known and predicted posttranslational modifications (ptms, Figure 1B) shows a high density of functional regions throughout the protein.

Variants from Bupp et al. and Rodan et al. [31,32] result in the early termination of the ODC C-terminal end, without disturbing the main enzyme structure. Our evolutionary data showed low conservation and selection for the C-terminal end of ODC1 throughout vertebrate evolution, but upon further investigation, we showed that outside of a few fish species, the C-terminal region is under high conservation (Appendix A). Amino acid 454 is a high probable S-farnesylation site (GPS-12.7 score), with the predicted modified cysteine under high conservation in 215/220 sequences analyzed, the most conserved site of all C-terminal amino acids. This amino acid is the only highly predicted lipidation site within ODC, with C202 having a score of 2.2 for S-palmitoylation. It should be noted that ODC has been shown to be lipidated within cells and this drives membrane localization [57], without a current elucidation of the residue driving this modification. Thus, it is highly probable that in addition to the C-terminal role in proteasome degradation [58], this region mutated in clinical cases may contribute to ODC lipidation and cell localization issues.

We built an integrated nonbiased genomic assessment (Figure 1C) within rare diseases (ClinVar, Geno2MP), population (TOPMed, gnomAD), and in cancer (COSMIC). ClinVar as of 9/2020 contains no ODC1 missense, nonsense, or frameshift submissions, and the ClinGen tools have yet to annotate the gene. However, Geno2MP has phenotypic associations for multiple rare missense variants within ODC1 (Appendix A). Of these variants, four (G84R, C202R, S195G, T285M) were predicted by multiple tools (PolyPhen2, Provean, SIFT) to be damaging to protein function, with 3/4 linked to brain, head, or neck phenotypes. The G84R variant occurred in 122 individuals out of the 17,928 total individuals of Geno2MP (0.68% of individuals).

To understand if this is enriched within the Geno2MP database, we assessed common population and large-scale sequencing-based variants for allele counts (Figure 1C) and allele frequencies within subpopulations (Appendix A). G84R (coded as rs138359527, HGVS NC_000002.12:g.10444500C > T) occurred within all of these databases and was the only common variant to have a predicted impact on protein function (Figure 1D). The variant was found at the highest allele frequency within South Asian individuals (0.8%, Figure 1E) specifically Gujarati Indians in Houston, TX (1.5%) and Punjabi in Lahore, Pakistan (1%, Appendix A). The overall allele frequency for G84R within gnomAD v2 was 0.23% and in TOPMed it was 0.18%. Using an average value of gnomAD and TOPMed (0.21%), we observed around a 3-fold enrichment of G84R within Geno2MP relative to population frequency (0.68/0.21 = 3.24).

Multiple lines of evidence point to a loss-of-function role for G84R. The tools PolyPhen2, Provean, SIFT, Condel, and align-GVGD predict the variant to be damaging. G84R had a CADD score of 23.5, suggesting the variant to be in the top 0.1% of impactful variants within the genome. Our developed evolutionary tools showed a score of 1.25/2 (Figure 1F), where G at this position was conserved in all 220 species analyzed and had a selection rate z-score based on codon usage of 0.5, suggesting that wobble occurred and yet no amino acid change happened throughout evolution. In total, this variant had an additive score of 5.25/6, where only 8.3% of ODC1 variants in totality had a score at this level, the majority of which occurred once within gnomAD (Appendix A). Amino acid 84 falls at the end of an α helix, where amino acid 85 is involved in helix capping and the downstream β sheet is centrally packed into the structural fold (Figure 1F). Conservation of amino acids mapped onto the ODC structure revealed highly conserved internal packing, with minimal surface exposed residues conserved (Figure 1G). Very interestingly, the G84 change to R had a high potential to interact with residues such as D424, which would stabilize the protein structure while impacting the dynamics of the protein (Figure 1H). Molecular dynamic simulations over 20 nanoseconds for the wild type ODC and G84R showed that an arginine resulted in extensive changes to the protein behavior, noted by an elevation of movement for the region around amino acid 84, stabilization of the C-terminal region centered at 424, and a resulting shift to overall protein loss of motion, as shown by stabilizing dynamic correlations of amino acids throughout the simulation (Appendix A). A functional enzyme assay for recombinant ODC WT or ODC G84R showed a 2.5-fold reduction of enzyme activity because of the variant, confirming alteration of enzymatic function due to the variant (Figure 1I). The G84R variant fell away from the antizyme binding site (Appendix A); however, we do not know the role of the change in ODC structural changes that could impact ODC-antizyme interaction that would also alter enzyme function. Further studies are needed to study G84R on antizyme biology.

Given the low allele frequency in the population (<1%), we do not anticipate that either GWAS or PheWAS statistical strategies would be able to identify high probable associations to biological traits for G84R (Appendix A). The issues of population trait mapping are further hindered by the fact that most populations used for these mapping are deficient in South Asian individuals, in whom we observed the highest frequency for G84R. Addressing all types of variants for ODC1 within Geno2MP including noncoding, it can be observed that the G84R variant is the most common occurring, yet it is the 16th ranked allele frequency for variants of ODC1 within Geno2MP (Appendix A). Phenotypes within the Geno2MP dataset for G84R (Appendix A) noted a high level of neurological traits including intellectual disability and microcephaly (Figure 1J). This supports that loss-of-function G84R has a phenotypic outcome for neurological impact. However, future in vivo analyses and co-segregation strategies should be performed to establish the causal roles of ODC G84R into phenotypic changes.

### 3.2. ODC1 Expression and Gene Regulation Variants

An assessment of expression from the Human Protein Atlas, GTEx, FANTOM, single cell expression databases, and the Allen Brain Atlas (Appendix A) showed high expression in tissues such as muscle for adult tissues. Coexpressed genes from single cell RNA-Seq of adult mouse tissues showed highly associated pathways of rRNA metabolic process, preribosome, and nuclear lumen. Focusing on the brain, based on observed neurodevelopmental phenotypes in rare diseases, expression of ODC1 was found in all regions (Appendix A) with a marked elevation correlating to early development, while neurons/glia progenitors are still proliferating and maturing (Figure 2A,B, Appendix A). Genes found highly correlated with ODC1 expression in the developing brain significantly enrich for cellular nitrogen compound metabolic process, nucleic acid metabolism, gene expression, nucleic acid binding, nucleus, and metabolism of RNA, all suggestive of involvement in cell proliferation (Appendix A). Only 17 genes had an R2 less than −0.9 in brain expression, with an enrichment of genes involved in myelin sheath and neuron projections (Appendix A). Of the 213 genes with 0.9 R2 to ODC1 expression in the Allen Brain Atlas, there were a total of 129 phenotype terms listed for pathogenic variants connected to brain/neurological connected disorders (Appendix A), further supporting a connection of ODC1 biology to early neurological formation.

To test the hypothesis that ODC1 expression correlates with the maturity of neurons, we developed induced pluripotent stem cells (iPSCs) differentiated into cerebral organoids followed by single cell RNA-Seq. These organoids represent the 12–16 pcw stage of development where ODC1 begins decreasing expression within the developing brain. We note a marked lack of expression of ODC1 on the edges of the UMAP for cerebral organoids, hallmarked by distance from the F group, which are the most immature cells (Figure 2C). There was also a higher expression of ODC1 in cells expressing pluripotent SOX gene markers, with both the number of cells and total expression levels of ODC1 lower in cells that did not have SOX gene expression (Figure 2D). This is interesting as SOX genes including SOX2 in the brain denote proliferative neural stem cells [59]. Consistent with this, Odc1 mRNA expression was elevated in distinct brain regions during mid-gestation of mouse brain using Genepaint [49] (Appendix A). Importantly, both medial and lateral regions of the brain had high levels of Odc1 in ventricular zones known to be highly prolific during these ages, suggesting a correlation between Odc1 expression levels and dividing brain cells. This evidence points to a critical window and location of ODC1 expression within the developing brain that correlates to gain-of-function variants resulting in macrocephaly (over proliferation of neurons) or loss-of-function variants to microcephaly (under proliferation of neurons).

The genomic landscape around ODC1 suggests complex quantitative trait loci (QTLs). A weakly significant but large effect size splicing QTL (sQTL) was observed for the 5′UTR of ODC1 within cultured fibroblasts, resulting in a protein of the same sequence (Appendix A). The highest probability came from rs11675020 (HGVS NC_000002.12:g.10528191G > C), a SNP specifically found in African decent, highest in Esan in Nigerian individuals (ESN, 35%). Very little is known about the role of the change in the 5′UTR of ODC1 and the involvement of these SNP in phenotypic associations due to the limited sample collections of African descent, representing a future minority characterization opportunity in ODC1 biology.

Multiple variants exist that are associated with changes in the expression levels of ODC1, as seen in broad tissue types (Figure 2E). Tissues including thyroid, tibial nerve, muscle, and lung show multiple association SNPs of various significance and normalized effect size (NES). These SNPs with eQTLs were taken through linkage disequilibrium (LD) analysis, with a cutoff of 0.8 R2, to impute linked SNPs. In total, we identified 900 variants connected to ODC1 changes in expression level (Figure 2F), over 1,414,872 bases around ODC1 (Appendix A). A total of 160 of these variants were connected to three different tissue eQTLs, showing that eQTL signatures overlap tissues. Integrating multiple levels of knowledge for each variant from CADD, ENCODE, Roadmap Epigenomics, RegulomeDB, SNP distances, GeneHancer, genome annotations, conserved transcription factor binding sites, Switchgear mapped promoters, and transcription factor binding alteration prediction, we developed a score for top functional regulation variants within each LD block. Three variants were found with high scores (Figure 2G), two of which (rs28742580 and rs2430422, HGVS NC_000002.12:g.10448426C > A and NC_000002.12:g.10447743G > A) were in LD and are the probable SNPs for the most significant (*p* = 1 × 10^−14^) ODC1 eQTLs seen from skeletal muscle only (Appendix A). The highest scoring SNP, rs2302615 (HGVS NC_000002.12:g.10448012C > T), had an eQTL for skeletal muscle (*p*-value 9.4 × 10^−7^) and was found over the top of the ODC1 gene body with occurrence in all populations, with East Asian at 57% (Appendix A). No significant PheWAS or GWAS have been connected to these SNPs to date, however, rs2302615 had some signal just below significance cutoff (*p* = 0.000062) for lung cancer from the UK biobank (https://genetics.opentargets.org/variant/2_10448012_C_T, accessed on 27 June 2020).

The observation of eQTL SNPs for tibial nerve (267, Figure 2E) further suggested a connection of ODC1 to neural biology. One of the SNPs on the eQTL list within tibial nerve, rs9287719 (Figure 2H, HGVS NC_000002.12:g.10570604C > T), had GWAS significance for prostate cancer (*p*-value 6 × 10^−17^ and 3 × 10^−8^) [50,51]. The lead SNP, which carries 161 LD SNPs found in most of the population, has three tissues of eQTLs for ODC1 (tibial nerve, skeletal muscle, lung) and is found within the NOL10 gene body, has structural looping and DNA hypersensitivity linking to a CpG island near ODC1, and has multiple SNPs of high scoring for regulation potential (Appendix A). From Open Target Genetics, this LD block was significantly connected (*p*-value 2.8 × 10^−7^) based on the UK biobank to the leg impedance. Within this region was also rs77575195 (HGVS NC_000002.12:g.10561470T > C), which had the highest eQTL significance of the region with 29 total SNPs in R2 > 0.8 (*p*-value 8.3 × 10^−9^, Figure 2I, Appendix A). Multiple tissue capture Hi-C and DNase hypersensitivity linkage analysis suggest chromosome looping of the NOL10 LD blocks to the ODC1 promoter (Figure 2J), further suggesting distal enhancer regulation. Overall, from eQTL analysis, we noted a small, normalized effect size on ODC1 expression from the 900 linked variants, suggesting a high selection on ODC1 expression levels throughout human evolution.

Many genomic regions loop to the ODC1 promoter, with a 1.4 million base pair region (chr2:9892007–1306879) linked to ODC1 regulation based on eQTLs. We assessed this region for rare variant phenotype enrichment using Geno2MP (Figure 2K). The SNPs linked to ODC1 eQTLs and ODC1 gene body region tended to have low HPO profile counts and CADD scores, except for ODC G84R. Of all SNPs within the 1.4 million base pair region, ODC G84R was an outlier for HPO profile counts and CADD score. Of the seven highest SNPs showing enrichment within Geno2MP with high CADD scores, there were 44 IEU Open GWAS project listed PheWAS hits including multiple psychiatric/neurological related traits (Appendix A). Out of the five PheWAS for G84R, three were psychiatric/neurological including the most significant (*p*-value 5.6e-4) with strong β (0.451), Right Long Insular Gyrus, and Central Sulcus of The insula-Surface Area (a2009s rh G Ins lg&S cent ins area). Overall, this suggests that out of all ODC1 connected genomic variants, ODC G84R was the most common neurological connected loss-of-function variant, with multiple lines of evidence for how such loss-of-function or gain-of-function variants and the ODC1 gene connect to neural development.

## 4. Discussion

Currently, around 100 publications have been curated for human *ODC1* (https://pubmed.ncbi.nlm.nih.gov/?from_uid=4953&linkname=gene_pubmed, accessed on 2 Februrary 2021) within PubMed, primarily focused on oncology. ODC is the entry enzyme for the synthesis of polyamines, connecting the enzyme to cell proliferation and transformation of cells [60]. This makes the *ODC1* gene a gatekeeper for multiple processes that underly both normal and disordered growth and development [61]. Like many oncogenes, proliferation control overlaps with early developmental processes, with germ line variants linked to dysmorphology including neurodevelopmental delay [62,63]. In these early developmental time points, proliferation control is critical. Within our work, we highlighted the expression of *ODC1* within critical early brain development that correlated to proliferation issues connecting loss-of-function variants, or inhibition, to decreasing proliferation control and microcephaly, while gain-of-function variants elevating proliferation connect to macrocephaly.

Our extensive expression analysis of neurological *ODC1* presented here agrees with noted human brain ODC levels elevated in the perinatal period that declined sharply during the first year of life and maintained throughout adulthood [64]. The over-abundance of ODC in Alzheimer’s disease postmortem samples have suggested a potential involvement in neurodegenerative diseases. High levels of polyamines have been connected to schizophrenia, mood disorders, anxiety, epilepsy, learning, and suicidal behavior [65]. The levels of ODC and polyamine activity are maintained in a fine balance in the central nervous system. Loss-of-function of ODC levels has been implicated in exacerbating cerebral occlusion damage and mortality [66], implicating that loss-of-function variants could hold risk to neurological damage.

Genes involved with the proliferation control of early neurons are normally deficient in variants impacting protein function or expression. For *ODC1*, we observed few noncoding variants linked to the large effect size of *ODC1* expression (eQTLs) and only a single missense variant of functional outcome (G84R). This one variant had the highest potential rare disease phenotypic risk (based on Geno2MP and CADD) within a 1.4 million base pair region around *ODC1*. Based on Bupp et al. and Rodan et al. [31,32], there is already a clear establishment of gain-of-function variants within *ODC1* to developmental delay, while inhibition of ODC with DFMO in early development impacts many biological pathways, suggesting a tight control of *ODC1* in development. The polyamine metabolites and genes connected to their processing have been connected to neurological dysfunction [30]. The linkage from the expression data of *ODC1* in early neural progenitor proliferation, to phenotypes of neurodevelopmental delay and brain area PheWAS, to the connection of polyamine metabolites in brain function establish *ODC1* as a bona fide neurodevelopmental disorder gene.

In the past few decades, few areas in science have gotten as much attention as genetics and genomics. Our ability to sequence and research genes, finding the next high impact variant that associates with disease states or ultra-rare variants that cause disorders is robust. However, we continue to identify new genes linked to rare disease phenotypes such as that of *ODC1*. So, what are we missing? Why do these genes and variants take a decade of sequencing to identify?

At least four aspects make genes such as *ODC1* difficult to establish neurodevelopmental delay association with our classical sequencing and statistical analysis. First, extreme changes such as dominant negative or complete loss of the gene are embryonic lethal, requiring more advanced tools to understand partial loss of the gene function and filtering of causal variants for disease. Second, variants in the 0.1 to 2% of populations are difficult to map phenotypic associations. In ultra-rare variant disease mapping, cutoffs for allele frequency remove these from possible candidates. In GWAS and PheWAS, these variants occur infrequently enough that they are unable to reach 1 × 10^−8^ genome-wide significance. Thus, we lack the mechanisms in our current tools to flag these for disease associations. Third, building on the GWAS, when variants occur in non-European ancestry, they often lack representation within phenotype association studies, further limiting statistical power in association identification. Finally, we still have a lot of genes, actually most, in which we do not fully understand their connections to diseases because of their complex pathways involved in intricate cell types and specific time points of development. Thus, we are just developing the tools to understand these details to connect genes to pathology, as laid out here for *ODC1*. Further complicating this is that genetics are not static, rather, toxicogenomic and environmental impacts influence genes of early development such as the role of testosterone or estrogens on neurological development [67,68], both of which are known regulators of *ODC1* [69,70,71].

Neurological disorders are not alone in these limitations of our current genomics, rather connected to nearly all phenotypes. Our work in *SHROOM3* for chronic kidney disease has shown that GWAS are underpowered for ethnically diverse variants in the ~0.5 to 1% range, requiring functional validation techniques and data integrations to decipher mechanisms to build disease association [72]. As presented here, we need to expand and integrate diverse datasets and filtering tools, which are increasing in power with techniques like single cell RNA-Seq, iPSC derived human organoids, the incredible resources of the Allen Brain Atlas, and other development tools sets.

Developing a in depth-knowledge of the ontology of the *ODC1* gene and its activity in different parts of the brain may prove to be a novel important target for the prevention and therapeutic treatment of these diseases that have a high impact on the costs of public health and considerably affect quality of life.

## 5. Conclusions

In addition to the recently discovered ODC gain-of-function C-terminal variants, ODC G84R (rs138359527) has statistical enrichment in patients with neurological conditions. Integrating this genomics with large scale expression of *ODC1* suggest a critical window of high expression of the gene during early brain development in locations connected to micro and macrocephaly. The linkage from the expression data of *ODC1* in early neural progenitor proliferation to phenotypes of neurodevelopmental delay and to the connection of polyamine metabolites in brain function establish *ODC1* as a bona fide neurodevelopmental disorder gene.

## Figures and Tables

**Figure 1 genes-12-00470-f001:**
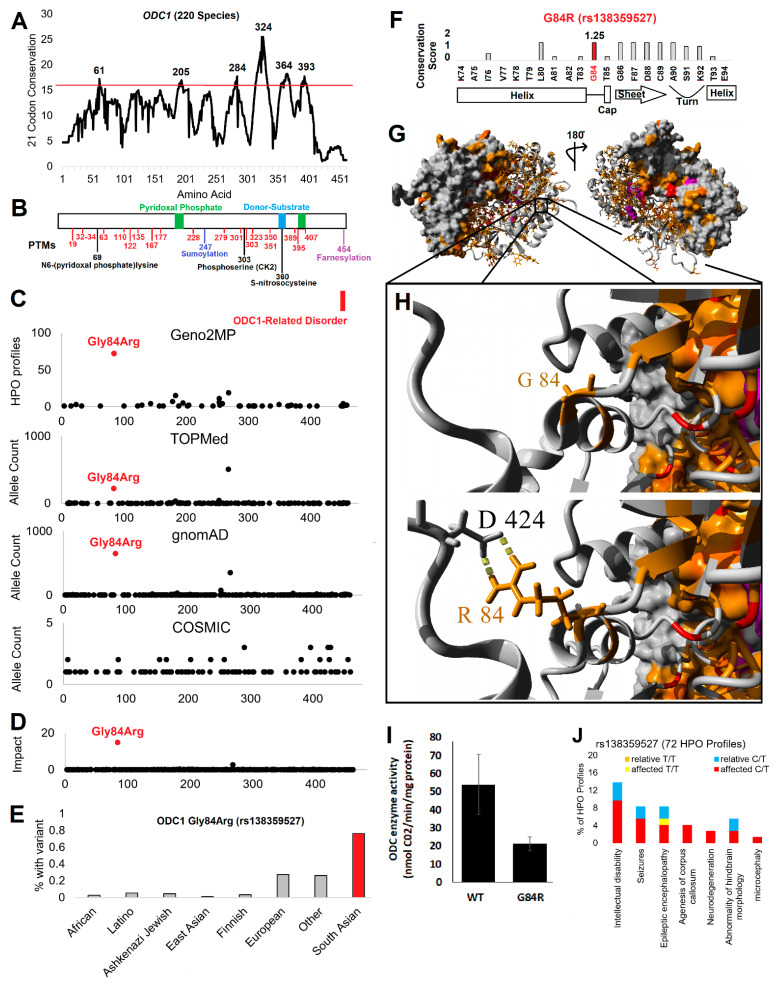
ODC1 missense variant screen. (**A**) Using 220 open reading frame sequences a 21-codon window for conservation was assessed for conserved motifs. The top six are labeled with the highest scored amino acid within the motif labeled. (**B**) Pictorial of ODC binding sites and posttranslational modifications. Known modifications are shown in black, predicted phosphorylation sites in red, sumoylation site in blue, and lipidation site in magenta. (**C**) Genomic variant extraction from the literature connected to ODC1-related disorder, Geno2MP, TOPmed, gnomAD, and COSMIC. The Geno2MP is shown as the number of patients with variant and human phenotype annotations (HPO profile). The G84R variant is labeled in red. (**D**) Combined variant impact scoring of all variants from panel C. (**E**) The allele frequency in various populations for G84R extracted from gnomAD. (**F**) Conservation score from 220 sequences for G84 region with a pictorial of secondary structure from the protein shown below. (**G**) Protein structure of ODC showing conserved amino acids (value >1.5 red, value >1 orange) and bound molecules in magenta. The dimer is shown with one unit as a surface plot and the other as the secondary structure. (**H**) Zoom in view of G84 (top) or energy minimized model of G84R. Colors are the same as panel H. (**I**) ODC enzyme activity. The enzymatic activities of purified human wild type (WT) ODC and mutated (G84R) ODC are compared side-by-side using the 14C ornithine labeled radioactive in vitro ODC assay. Data represent four independent experiments with multiple technical replicates (*n* = 10) ± standard deviation. *p* < 0.001. (**J**) Individuals HPO terms for G84R. Colors correspond to homozygous or heterozygous status and whether individual was an affected patient or their relative.

**Figure 2 genes-12-00470-f002:**
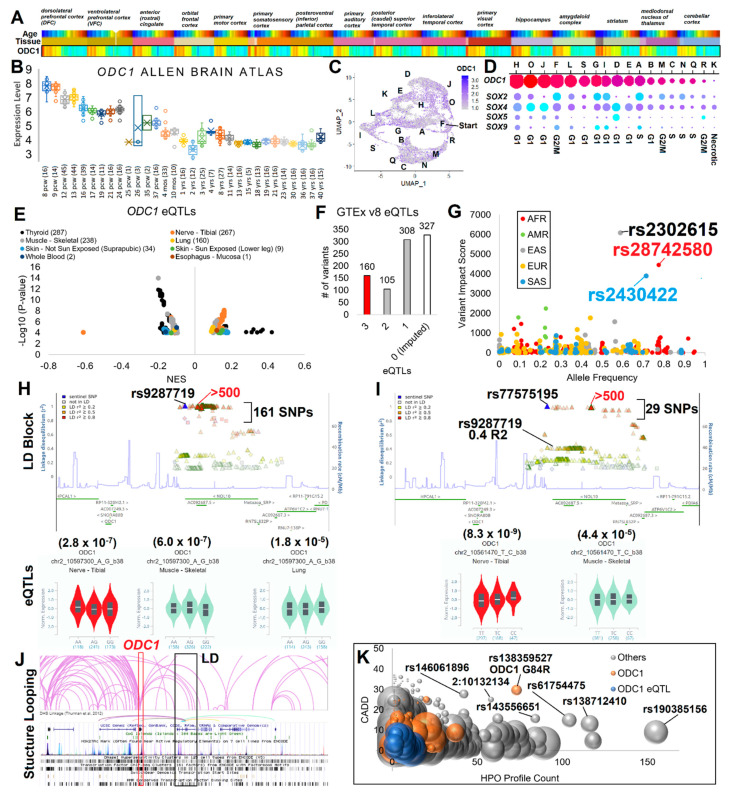
ODC1 gene regulation and functional genomics. 10^−7^ (**A**) Allen Brain Atlas Developmental Transcriptome data for ODC1 expression. On top are labeled tissue types. First color heatmap is for age, with youngest in blue and oldest samples in red. ODC1 levels are on a color scale of red (high) to blue (low). (**B**) Clustering of ODC1 expression from panel A for each age group shown as a box and whisker plot. The number of tissues and samples are marked in brackets for each age group. (**C**) UMAP for cerebral organoid single cell RNA-Seq showing ODC1 expression. Cell cluster annotations are labeled. (**D**) Expression of ODC1 in each cluster is shown as a dot plot. The size of each spot represents the percent of cells in the cluster expressing ODC1 and the color corresponds to the expression levels. Shown below are dot plots for four SOX genes. (**E**) Volcano plot of ODC1 linked eQTLs with tissue types colored and legend labeling number of eQTLs observed in each. (**F**) The number of times eQTLs (or 0.8R2 imputed) occurred across different tissues. (**G**) SNPs shown for allele frequency (x-axis) vs. combined variant impact score (y-axis). Colors correspond to the population with the highest frequency. (**H**,**I**) Linkage disequilibrium plot for top two tibial nerve eQTL regions. SNPs called out in red have impact scores above 500 and the lead eQTL SNP is labeled in blue. Shown below is the volcano plots for lead SNPs in different tissues (tibial nerve in red) with *p*-value shown above. (**J**) Hi-C lopping (lung tissue) and DNA hypersensitivity linkage from ENCODE cell lines for the LD regions in panel H-I looping to ODC1. (**K**) Bubble plot for SNPs within the region of ODC1 eQTLs extracted from Gene2MP. X-axis shows the number of HPO profiles (patients with phenotype information), the y-axis shows the CADD score of each SNP, and the size of the bubble corresponds to scaled allele frequency in gnomAD. Bubbles in orange are within the ODC1 gene region, those in blue are ODC1 eQTLs, and those in gray found throughout this ODC1 region.

## Data Availability

Appendix A contain the mentioned data. All single cell RNA-Seq is available as BioProject PRJNA640661.

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
