# Peer review of "Emerging Role of ODC1 in Neurodevelopmental Disorders and Brain Development"

_genes, 2021, doi:10.3390/genes12040470_

Round 1

Reviewer 1 Report

Jeremy and colleagues found a loss-of-function variant in ODC1 associated with intellectual disability and seizure by analyzing various database and revealed a genotype-to-phenotype associations in ODC1 by using Expression quantitative trait loci (eQTLs), coupled with eQTLs enzyme activity and RNAseq of cerebral organoids. The work is carefully done and well-described. The data are of high quality and the manuscript is well-written. Specific comments are below.

Materials and Methods.

“Each missense variant was analyzed with PolyPhen2 (probably=1, possibly=0.5), Provean (Deleterious=1), SIFT (Damaging=1), align-GVGD (Class C65/55=1, C45/35=0.5, C25/15=0.25), and our conservation score (max=2) with a summed max variant score of 6.This variant score was multiplied by the 21-codon conservation and gnomAD allele frequency for a combined impact score.”

Is there a standard for the impact score or why you choose such a combination of tools to assess the variant? Given there are a lot of missense variant prediction tools with different criteria, such as PolyPhen2 and align-GVGD are based on protein structure/function and evolutionary conservation, SIFT is based on evolutionary conservation and Provean is based on alignment and measurement of similarity between variant sequence and protein sequence homolog.

Similar question in the “2.1 Coding variant screening” part, “A compiled noncoding variant impact score was generated by summing the number of overlapping peaks from ENCODE, Roadmap, RegulomeDB, and GeneHancer and multiplying by CADD score”.

We pretty sure that the levels of ODC and polyamine activity are maintained in a fine balance in the CNS. However, the variants in 1% of populations is slightly high to make definite phenotypic associations, even there have some evidence for functional modify. More functional validation and co-segregation analysis is needed.

Author Response

“Each missense variant was analyzed with PolyPhen2 (probably=1, possibly=0.5), Provean (Deleterious=1), SIFT (Damaging=1), align-GVGD (Class C65/55=1, C45/35=0.5, C25/15=0.25), and our conservation score (max=2) with a summed max variant score of 6.This variant score was multiplied by the 21-codon conservation and gnomAD allele frequency for a combined impact score.”

Is there a standard for the impact score or why you choose such a combination of tools to assess the variant? Given there are a lot of missense variant prediction tools with different criteria, such as PolyPhen2 and align-GVGD are based on protein structure/function and evolutionary conservation, SIFT is based on evolutionary conservation and Provean is based on alignment and measurement of similarity between variant sequence and protein sequence homolog.

Response: We added, “This combined score allows for ranking of variants that are predicted to impact function from multiple tools, prioritizing variants.”

Similar question in the “2.1 Coding variant screening” part, “A compiled noncoding variant impact score was generated by summing the number of overlapping peaks from ENCODE, Roadmap, RegulomeDB, and GeneHancer and multiplying by CADD score”.

Response: We added, “This score allows for the rapid screening of any variants where epigenetic knowledge is known and where there might be overlapping signals.”

We pretty sure that the levels of ODC and polyamine activity are maintained in a fine balance in the CNS. However, the variants in 1% of populations is slightly high to make definite phenotypic associations, even there have some evidence for functional modify. More functional validation and co-segregation analysis is needed.

Response: We added, “However, future in vivo analyses and co-segregation strategies should be performed to establish the causal roles of ODC G84R into phenotypic changes.”

Reviewer 2 Report

This work aimed to demonstrate the association of ODC1 genetic and mRNA variants with neurological disorders and brain development. The authors have done a comprehensive data analysis and computer modelling. They find that the most common neurological-associated variant is G84R, enriched in South Asian population. In addition, they demonstrate that the recombinant G84R variant displays lower ODC activity in vitro. The study is well-conducted, and results support the conclusions made.

Comments:

ODC1 (and many other key genes of polyamine metabolism) is regulated mainly at posttranslational level. The half-life and activity of ODC1 enzyme protein is regulated by antizyme proteins, and antizymes in turn are regulated by antizyme inhibitor proteins. I recommend adding this background to the text (introduction or discussion), since the average reader may not be aware of this kind of complex regulation.

I wonder, how the G84R missense mutation affects the binding of antizyme to ODC? Could the authors briefly discuss their thoughts on this issue?

Minor comment:

Acknowledgments-section has not been filled.

Author Response

ODC1 (and many other key genes of polyamine metabolism) is regulated mainly at posttranslational level. The half-life and activity of ODC1 enzyme protein is regulated by antizyme proteins, and antizymes in turn are regulated by antizyme inhibitor proteins. I recommend adding this background to the text (introduction or discussion), since the average reader may not be aware of this kind of complex regulation.

Response: We added, “The regulation of polyamines and ODC activity is critical. This importance can be high-lighted by the evolved role of ODC antizyme and antizyme inhibitors [21]. The ODC an-tizyme inhibits ODC and its functional role is highly dependent on the level of polyam-ines in the cell [22–24]. The unique frameshift regulation of antizyme and role in degra-dation of ODC makes the antizyme system a novel regulatory component [25]. Adding to the complexity of this regulatory system is a homologous ODC1 that can inhibit the an-tizyme from ODC regulation [26,27].”

I wonder, how the G84R missense mutation affects the binding of antizyme to ODC? Could the authors briefly discuss their thoughts on this issue?

Response: We added a new figure S6 and added, “The G84R variant falls away from the antizyme binding site (Supplemental Figure S6); however, we do not know the role of the change of ODC structural changes that could impact ODC-antizyme interaction that would also alter enzyme function. Further studies are needed to study G84R on antizyme biology.”

Minor comment:

Acknowledgments-section has not been filled.

Response: The section has been removed.